# Nanotechnology in Bladder Cancer: Diagnosis and Treatment

**DOI:** 10.3390/cancers13092214

**Published:** 2021-05-05

**Authors:** Mahmood Barani, Seyedeh Maryam Hosseinikhah, Abbas Rahdar, Leila Farhoudi, Rabia Arshad, Magali Cucchiarini, Sadanand Pandey

**Affiliations:** 1Department of Chemistry, Shahid Bahonar University of Kerman, Kerman 76169-14111, Iran; mahmoodbarani7@gmail.com; 2Nanotechnology Research Center, Pharmaceutical Technology Institute, Mashhad University of Medical Sciences, Mashhad 91886-17871, Iran; Hoseinikhm961@mums.ac.ir (S.M.H.); FarhoudiL961@mums.ac.ir (L.F.); 3Department of Physics, Faculty of Science, University of Zabol, Zabol 98613-35856, Iran; 4Department of Pharmacy, Quaid-I-Azam University, Islamabad 45320, Pakistan; rabia.arshad@bs.qau.edu.pk; 5Center of Experimental Orthopaedics, Saarland University Medical Center, 66421 Homburg/Saar, Germany; 6Department of Chemistry, College of Natural Science, Yeungnam University, 280 Daehak-Ro, Gyeongsan 38541, Korea; 7Particulate Matter Research Center, Research Institute of Industrial Science & Technology (RIST), 187-12, Geumho-ro, Gwangyang-si 57801, Korea

**Keywords:** bladder cancer, nanoparticles, therapy, diagnosis

## Abstract

**Simple Summary:**

Bladder cancer (BC) is the fourth most common cancer among men and the tenth most common cancer among women. Since the overall prognosis for BC has not changed in the last 30 years, there is a compelling medical need to develop new diagnostic and therapeutic approaches. Nanotechnology has been extensively developed for cancer management, including cancer diagnosis, detection, and treatment. Several nanoparticles (NP) can be used in in vitro cancer diagnostics, in vivo imaging enhancement, and drug loading techniques. In this review, we examine the current state of nanotechnology in the diagnosis and treatment of bladder cancer. We investigated the function of metal NPs, polymeric NPs, liposomes, and exosomes accompanied therapeutic agents for BC therapy, and then focused on the potential of nanotechnology to improve conventional approaches in sensing.

**Abstract:**

Bladder cancer (BC) is the second most common cancer of the urinary tract in men and the fourth most common cancer in women, and its incidence rises with age. There are many conventional methods for diagnosis and treatment of BC. There are some current biomarkers and clinical tests for the diagnosis and treatment of BC. For example, radiotherapy combined with chemotherapy and surgical, but residual tumor cells mostly cause tumor recurrence. In addition, chemotherapy after transurethral resection causes high side effects, and lack of selectivity, and low sensitivity in sensing. Therefore, it is essential to improve new procedures for the diagnosis and treatment of BC. Nanotechnology has recently sparked an interest in a variety of areas, including medicine, chemistry, physics, and biology. Nanoparticles (NP) have been used in tumor therapies as appropriate tools for enhancing drug delivery efficacy and enabling therapeutic performance. It is noteworthy, nanomaterial could be reduced the limitation of conventional cancer diagnosis and treatments. Since, the major disadvantages of therapeutic drugs are their insolubility in an aqueous solvent, for instance, paclitaxel (PTX) is one of the important therapeutic agents utilized to treating BC, due to its ability to prevent cancer cell growth. However, its major problem is the poor solubility, which has confirmed to be a challenge when improving stable formulations for BC treatment. In order to reduce this challenge, anti-cancer drugs can be loaded into NPs that can improve water solubility. In our review, we state several nanosystem, which can effective and useful for the diagnosis, treatment of BC. We investigate the function of metal NPs, polymeric NPs, liposomes, and exosomes accompanied therapeutic agents for BC Therapy, and then focused on the potential of nanotechnology to improve conventional approaches in sensing.

## 1. Introduction

The bladder is an essential organ of the urinary system having major roles in temporary urine storage via immense folded internal linings, as well as its expulsion due to bladder musculature contractions and relaxations [1,2]. The bladder is featured with the apex at the upper location, main body, triangular-shaped posterior fundus, as well as neck for convergence of fundus [3]. BC is the common cancer of the urinary tract, originating inside the linings of the bladder consisting of urothelial cells [4]. Urothelial cells are the source of connection between the kidney and bladder [5]. BC is the fourth most common cancer in men and tenth in women causing 15,000 deaths annually in the USA [6]. BC includes a variety of forms, depending on how it develops in the bladder’s specific cells, i.e., urothelial carcinoma, squamous cell carcinoma, and very rare adenocarcinoma in mucus-secreting cells of the bladder [7]. As far as symptoms of BC are concerned, it is often accompanied by hematuria, frequent painful urination, and pelvic pain [8]. The mechanism besides the propagation of BC lies in the resistant mutations in the tumor suppressor genes [9]. However, other risk factors include smoking, old age, chronic bladder inflammation, inheritance history, and exposure to certain chemicals and dyes [4,10,11,12,13]. As far as a generalized diagnosis of the BC is concerned, it can be initially determined through cystoscopy by inserting a cystoscope tube having a lens into the urethra to examine structural changes. Cystoscopy can be modified via attaching a specialized tool for collecting a sample for biopsy determination. Furthermore, the urine sample can also be opted to collect for analysis of the cancer cells presence [14,15]. After the confirmation of the presence of traces of cancer cells in urine confirming BC. Further preceded diagnostics can also be performed to examine the severity, and it includes CT scan, magnetic resonance imaging (MRI), positron emission tomography (PET), bone scan, and chest X-ray. The diagnosis of BC is challenging owing to the vast existence of gaps accompanying over-testing, over-diagnosis, over-treatment, non-specificity, and heterogeneous nature of malignant BC cells [16]. However, treatment is varied according to the type and severity based on low grade (benign) and high grade (metastatic) BC [8,17]. Therefore, the opted methods for BC therapy include surgery for removing cancer cells directly via transurethral resection of bladder tumor (TURBT), cystectomy, neobladder reconstruction, and ileal conduit [18,19]. Chemotherapy for BC includes the use of combinatorial chemotherapeutic agents in the form of gemcitabine and cisplatin (GC), cisplatin, methotrexate, and vinblastine (CMV), and gemcitabine and paclitaxel (GP) directly in the bladder via intravesical chemotherapy or systemic chemotherapy [17,20,21]. Radiation therapy uses beams of powerful energy, such as X-rays and protons, to destroy the cancer cells [22]. Immunotherapy is the most used treatment protocol for BC via intravesical and intravenous methods [23]. Intravesical immunotherapy can be done using the *bacillus Calmette-Guerin (BCG) vaccine* to mediate the immune system reaction that directs germ-fighting cells to the bladder [24]. However, intravenous immunotherapy can be performed through various immunotherapy drugs [25]. Nevertheless, all these modalities methods are associated with an increased economic burden, non-patient compliance, and need for targeted delivery, and most importantly with the involvement of virulent factors in tumor suppressor genes [26]. Chemotherapeutics agents utilized in chemotherapy are specifically associated with damaging cells in the bone marrow, intestinal mucosal linings and hair follicles, as well as developing severe infection and fatigue due to the depletion of leukocytes and red blood cells [27]. In terms of a generalized diagnosis of BC, it can be determined first by cystoscopy, which involves inserting a cystoscope tube with a lens into the urethra to investigate structural changes [28]. Cystoscopy can be modified via attaching specialized tool for collecting sample for biopsy determination [29,30]. Furthermore, urine sample can also be opted to collect for analysis of the cancer cells presence. After the confirmation of the presence of traces of cancer cells in urine confirming BC, further preceded diagnostics can also be performed to examine the severity and it includes CT scan, magnetic resonance imaging (MRI), positron emission tomography (PET), bone scan and chest X-ray [31,32,33]. The diagnosis of BC is challenging owing to the vast existence of gaps accompanying over-testing, over-diagnosis, over-treatment, non-specificity, and heterogeneous nature of malignant BC cells [34]. To overcome the limitations of conventional diagnostic methods for BC, a significant number of nanotechnology-based bioassays are highly encouraged [35]. In this regard, fluorescent cystoscopy has been developed using 5 -aminolevulinic acid (5-ALA) phostosensitizer followed by intravesical administration, and have the capability of proficient absorption of cancer cells by showing intense red color compared to surrounding normal tissues [36]. Similarly, ligand mediated approach-based nano-sensors are also of great interest as they can be developed by conjugating BC specified amino acid-based ligand PLZ4. PLZ4 functionalized nanomicelles preferentially enhance the uptake of only cancer cells under the mechanistic of photodynamic diagnosis [37]. Role of gold nanoparticles (GNP) in the diagnosis of BCs by inducing plasmon resonance is irreplaceable and resulting in colors of visible difference to be observed by naked eyes [38]. Cationic GNPs can be utilized with biocompatible anionic hyaluronic acid (HA) to produce visible blue color change [39]. Ultra-small particles of iron oxide (USPIO) ranging from 30 to 50 nm in diameter have the capability of uptake by reticuloendothelial system for ultra-sensitive diagnosis [40].

Nanotechnology advents the field of anti-cancer modalities via improving the drug loading by decorating the surface of nanoparticles with targeted receptor ligands, highly expressed on tumor surfaces [41,42,43,44,45,46,47,48]. Nanomaterials bypass the side effects of conventional therapy by improving the specificity and pharmacokinetics of anti-cancer drugs [49,50,51,52]. Anti-cancer NPs for treating BC are GNPs and they modified the acids and protein molecules for facilitating the rapid killing of cancer cells [53]. GNPs help in providing stability, as well as a strong affinity for attachment of ligands targeting BC [54]. Most effectively utilized nanocarriers in the treatment of BC include, polymeric nanoparticles because their synthesis is easy and cost-effective, provides superior viability and biodegradation [55]. Polymeric nanoparticles utilized in the treatment of BC are available in a wide range of natural and synthetic polymers, constituting macromolecules poly (lactide-coglycolide), poly (lactic acid), poly (caprolactone), and chitosan [55,56,57,58,59,60,61,62]. Similarly, the lipid-based nanoparticles are also imparting their role in treating BC by adapting formulation strategic forms of solid lipid nanoparticles (SLNs) and nanostructured lipid carriers (NLC) composed of phospholipids [63]. Lipid-based nanoparticles are highly advantageous in encapsulating anti-cancer drugs and their site-specific loading via overcoming their solubility issues [64]. Moreover, protein nanoparticles help in facilitating the targeted and controlled release of oral delivery using proteins such as albumin, gelatin, gliadin, and legumin [65].

## 2. Diagnosis of Bladder Cancer

### 2.1. Current Diagnosis Approaches of Bladder Cancer

Currently, different approaches such as cystoscopy, biopsy, urinary cytology, and imaging procedures are used to detect BC [66]. A thin, narrow tube (cystoscope) is placed through the urethra to perform cystoscopy. The cystoscope has a lens that helps to inspect inside of the urethra and bladder for signs of disease [67]. Urine cytology is a process in which a sample of urine is examined under a microscope to monitor for cancer problems [68]. Finally, imaging examinations such as computerized tomography (CT) urogram, magnetic resonance imaging (MRI), positron emission tomography (PET), bone scan, and chest X-ray can help with early diagnosis and influence care decisions [69].

Traditional methods have limited sensitivity and capacity to provide accurate and specific information on the condition, and they rely greatly on the size shift of lymph nodes and the presence of anatomic anomalies, which is typically the primary criteria used to determine the diagnosis [29,70]. Nanoparticles (NP) with a nanometric scale are novel materials that were first used to develop scanning probe microscopy and discover molecular structures, and since been used in a variety of diagnostic applications [71].

### 2.2. Current Biomarkers and Tests in Bladder Cancer Diagnosis

The identification of biomarkers in urine, tissue, and blood has been suggested as important components of precision medicine to address existing shortcomings in the diagnosis, treatment, and follow-up of BC. These modern molecular tests can aid in detecting disease earlier, risk-stratifying patients, improving oncological outcome prediction, and optimizing target therapies. We looked at the existing state of the art and the success of the most promising and accessible biomarkers. BTA stat^®^, BTA TRAK^®^, NMP22, and UroVysionTM are the only urinary diagnosis systems approved by the Food and Drug Administration (FDA) for BC diagnosis and follow-up. Meanwhile, *ImmunoCytTM/uCyt+TM* is only approved for BC follow-up [72]. Telomerase, nuclear matrix protein 22 (NMP22), cytokeratin 19, survivin, hyaluronidase (HAase), apolipoprotein A1 (ApoA1), miRNA-21, and galectin-1 protein, on the other hand, are effective biomarkers in the diagnosis of BC [73,74].

Despite some promising findings, particularly in terms of improved sensitivity, when compared to urinary cytology in the diagnostic environment. These molecules have yet to be incorporated into routine clinical practice, owing to the difficulty in deciding the required scenario for use and the lack of high-quality prospective trials. That resulting in a low level of proof. To define the position of these promising biomarkers, more prospective research and broad international collaborations are needed [75]. Nanotechnology has been extensively developed for cancer management, including cancer diagnosis, identification, and treatment. Nanoparticles for cancer treatment have a plethora of potential uses, but most are still in the preclinical stage. GNPs, for example, would be used in cancer screening in vitro studies, in vivo imaging improvement, and as potential drug loading strategies. Aside from that, different preparation methods alter particle sizes, shapes, and structures, reducing their clinical utility [76,77,78,79].

### 2.3. Nanomaterials for Imaging Approach

Current imaging methods, such as computed tomography and magnetic resonance imaging (MRI), rely heavily on the size shift of lymph nodes or the presence of anatomic anomalies as the primary criteria for diagnosis [80]. Nanoplatform of contrast agents has shown some promising results in imaging approaches to address this problem [81].

BC, as previously mentioned, has the highest recurrence rate of all cancers, owing in part to insufficient transurethral resection. The failure of cystoscopes to identify invisible lesions during the resection process results in inadequate resection [82]. Davis et al evaluated the ability of the endoscope approach and surface-enhanced Raman NPs to detect bladder tissue as cancerous or normal (Figure 1) [83]. After topically administration of NPs to human bladder tissue samples, both tissue permeability-based (passive) targeting and antibody-based (active) targeting was assayed. The receiver operating characteristic region under the curve (ROC AUC), for multiplexed molecular imaging of CD47 and Carbonic Anhydrase 9 tumor proteins was 0.93. (0.75, 1.00). Furthermore, with a ROC AUC of 0.93, passively targeted NPs allowed tissue classification. As compared to standard bladder urothelium, passively targeted nanoparticles penetrated five-time deeper and attached to tumor tissue at 3.3-time higher concentrations in cancer. Indicating that there is an improved surface permeation and retaining function in human BC.

Quantum dots (QD) are fluorescent nanoparticles with superior optical properties compared to organic dyes, but in vivo use of them is constrained by the potential for systemic toxicity. Topical administration of targeted nanoparticles is appealing, because it allows for minimal exposure and dose reduction [84]. Pan et al. successfully indicated that topical (i.e., intravesical) administration of QD-conjugated anti-CD47 resulted in effective ex vivo endoscopic imaging of human BC. In addition, they investigated the biodistribution and toxicity of free QD, and anti-CD47-QD in mice in vivo [85]. Anti-CD47-QD biodistribution in vivo was measured using inductively coupled plasma mass spectrometry. While some mice showed extravesical biodistribution of QD, indicating a route for systemic exposure under some conditions, there was no substantial aggregation of QD outside of the bladder on average. Up to 7 days after injection, there were no signs of high toxicity. Intravesical administering of targeted NPs can minimize systemic exposure, but for clinical use, NPs with well-established bio - safety characteristics should be used to reduce long-term contamination in situations of drug toxicity.

The use of optical imaging technology in conjunction with cancer-specific molecular imaging agents may be an effective solution for promoting cancer detection and enabling image-guided surgical [86]. The most common endoscopic treatment for BC is white light cystoscopy, which has a low detection ability. Developing optical imaging innovations have a lot of promise for better diagnostic precision, but there are not any imaging agents for molecular specificity yet [87]. Pan et al demonstrated strong detection of BC with clinical-grade fluorescence imaging approaches, confocal endomicroscopy, and blue light cystoscopy in fresh surgically extracted human bladders using fluorescently labeled CD47 antibody (anti-CD47) as a molecular diagnostic agent [88]. The specificity and sensitivity for CD47-targeted imaging with blue light cystoscopy were 82.9% and 90.5%, respectively. Based on the results of this study, CD47-targeted molecular imaging could help with BC diagnosis and resection precision.

Despite numerous resections and long-term chemo and immunotherapy, most patients with non-muscle invasive BC undergo recurrence or worsening, necessitating cystectomy and a poor prognosis. Unfinished resection and reimplantation of cancer cells are potential causes, which could be avoided with better resection and adjuvant treatment [89]. Liang et al produced a guided drug for urothelial carcinoma identification, fluorescence-guided resection, and deep-penetrating adjuvant photodynamic therapy (UC) [90]. The method was based on up-conversion nanoparticles (UCNP), which can hold a photosensitizer and convert deep-penetrating near-infrared light into high-energy visible light, which is needed for tumor interpretation as well as the development of reactive oxygen species in the photosensitizer. The mean photoluminescence of cells in the targeted group was 5 to 8 times higher than in control groups as a result of the labeling, makes it for quick detection of positive cells with low background auto-fluorescence.

QDs are commonly used in biomedical fields as fluorescent labels. Yuan et al developed quantum dot (QD) fluorescent probes that were linked to a monoclonal antibody against prostate stem cell antigen (PSCA) (QD-PSCA). The targeted imaging of QD-PSCA probes in EJ human bladder urothelial cancer cells was investigated using this nanoplatform. They also looked into the efficacy of non-invasive tumor-targeted imaging using these QDs in vivo conditions. By chemical covalent linking, QDs with an emission wavelength of 605 nm (QD605) was covalently linked with PSCA to produce QD605-PSCA fluorescent probes. An ultraviolet spectrophotometer and a fluorescence spectrophotometer were used to test and determine the optical properties of the probes coupled and uncoupled with PSCA. To identify and interpret imaging of the probes for EJ cells, direct immune-fluorescent labeling was used. The findings showed that QD605-PSCA probes maintained QD605’s fluorescent properties, as well as the PSCA protein’s immunogenicity. The probes were able to identify the PSCA protein expressed in BC cells with high specificity, and the fluorescence was stable and long lasting. QD-PSCA fluorescent probes may be helpful for precise targeted labeling and imaging in bladder urothelial cancer cells, according to this report. The probes also have good optical durability, making them suitable for non-invasive targeted imaging, early detection, and targeted in vivo tumor therapy testing.

### 2.4. Nanomaterials in Biosensor Development

Researchers identified promising urinary biomarkers for the detection of BC as a result of their efforts to reduce the costs of BC diagnosis and improve patients’ quality of life by removing unnecessarily invasive diagnostic tests [91,92,93,94,95,96,97]. The potential biomarker for early diagnosis of BC is telomerase, nuclear matrix protein 22 (NMP22), cytokeratin 19, survivin, hyaluronidase (HAase), apolipoprotein A1 (ApoA1), miRNA-21, galectin-1 protein, etc. [98]. Biosensors for identifying urinary biomarkers have been established in tandem with their discovery, and they can give low detection limits, a large linear response range, high stability, and high accuracy [99].

#### 2.4.1. Gold Nanoparticle

Survivin is one of the biomarkers used to diagnose BC [100]. Using the ELISA approach and gold nanoparticle (GNP) covalently attached to survivin antibody, Jazayeri et al described reactive antibodies to survivin protein as a biomarker to define the BC phase [38]. The plasma and urine concentrations of survivin increased dramatically in the T3 and T4 phases of the disease (high grades), according to ELISA data, as compared to healthy people. Survivin protein was also found in the urine samples of patients of all classes using cross-linked GNPs (low and high grades).

Another urinary biomarker of BC has been identified as hyaluronidase (HAase) [101]. Nossier et al. developed a simple colorimetric GNPs assay for detecting urinary HAase activity quickly and accurately [102]. The formation of gold structures and a red to blue color change came from charge interaction between cationic CTAB-covered gold NPs and polyanionic hyaluronic acid (HA). In addition, poly-cationic chitosan was used to collect all negatively charged compounds in urine. In a study of 40 bladder carcinoma patients, 11 benign bladder lesions patients, and 15 healthy people, the developed GNP technique was compared to zymography for rapid identification of urinary HAase activity. The assay sensitivity was 82.5% vs. 65% for zymography, and the specificity for both tests was 96.1%. The HAase activity was measured using the A530/A620 absorption ratio of the reacted GNP solution. The highest cut-off achieved about 93.5 U/ng protein with 81% specificity and 90% sensitivity.

#### 2.4.2. Graphene

Peng et al developed a magnetic graphene oxide (GO)-linked to Prussian blue (PB) (PMGO) as a peroxidase-mimicking nanozyme with great oxidizability to 3,3′,5,5′-tetramethylbenzidine (TMB). For colorimetric immunosensing of apolipoprotein A1 (ApoA1) as a biomarker of BC, this probe provides considerable absorption capacity [103]. Immunosensor biochip was functionalized with ApoA1 antibody (AbApoA1) and PMGO. In the existence of self-linkable PMGO, the linear detection range was significantly expanded (from 0.05 to 100 ng/mL) as compared to the group without signal enhancement (from 1 to 100 ng/mL). The immunosensor evaluation of ApoA1 in the urine of BC patients and healthy people was strongly associated with enzyme-linked immunosorbent assay measurements; however, ApoA1 concentrations in high-grade BC patients were markedly larger than in low-grade BC patients.

In medical and biological science, molecular beacon (MB)-sensors have a high impact on proper diagnosis. The synthesis of both quencher and fluorogen in nucleic acid probes, would increase the strain of organic synthesis work and create difficulties in precisely regulating the relative distance between quencher and fluorogen, possibly leading to false-positive and false-negative outcomes [104]. Ou et al. published a single labeled MB (FAM-MB, with carboxyfluorescein as fluorogen and no quencher) to detect telomerase activity using GO [104]. They design an easy, responsive, and selective system using a label-free beacon (AIE-MB) based on accumulation emission fluorogen (silole-R), to further simplify this design, namely label-free strategy. The AIE-MB mediated comb-like DNA structure results in higher aggregation of silole-R and, thus, heavy fluorescence emission when telomerase is added. They can identify telomerase with greater sensitivity as a result of this, and show how it can be used to diagnose BC.

Duan et al used a system to analyze miRNA-21 and telomerase by a nicking enzyme-assisted signal amplification and GO [105]. The use of a DNA molecular beacon probe to prevent the formation of G-quadruplexes boosts telomerase activity. The implementation of GO greatly reduces background noise. This binary assay can differentiate between urine from BC patients, cystitis patients, and healthy people, according to tests on 258 urine samples. Eventually, this technique has a lot of promise when it comes to discriminating between muscle-invasive and non-muscle-invasive BCs.

#### 2.4.3. Inorganic Nanoparticles

Galectin-1 protein has recently been identified as a useful urinary biomarker for BC diagnosis and prognosis [106]. Shaikh et al. proposed a responsive and precise impedimetric immunosensor for detecting Galectin-1 protein in clinical urine samples quantitatively and without the use of labels. In total, nine gold interdigitated microelectrodes (3 × 3 array) make up the immunosensor [106]. They used Galectin-1/Al2O3 nanoprobes (Galectin-1 antibody bonded to alumina NPs) that can be specifically trapping on the microelectrode surface using positive dielectrophoresis (p-DEP) to achieve higher sensitivities. The median normalized impedance difference during immunosensing for 22 cancer patients and 26 normal patients is 27% and 10%, respectively, according to the clinical tests, with a cut-off point of 19.5% above which the specificity and sensitivity of BC diagnosis were 80% and 82%, respectively.

The enhancement of detection precision and specificity has been hindered by high background noise caused by contaminants and other analytes in biological mixtures. Wang et al developed an ultralow background bio-chip based on time-gated luminescent probes assisted by photonic crystals (PC) for high specificity and sensitivity identification of BC-related miRNA biomarkers in urine samples [107]. Auto-fluorescence can be efficiently eliminated when combined with the luminescence-enhanced capability of PCs and the time-gated luminescence of long-lifetime luminescence platforms. Thus, detection sensitivity would be dramatically improved. Taking advantage of these advantages, a detection limit of 26.3 fM was obtained. Moreover, the biochip performs well in detecting urinary miRNA, with strong recoveries. The produced biochip has ultralow background and luminescence enhancement capabilities, making it an excellent tool for detecting BC-related miRNA in urine.

#### 2.4.4. Extracellular Vesicles (EV)

Extracellular vesicles (EV), which include microvesicles and exosomes, can be found in the human body, and the concentration of exosomes and their related biomarkers such as nucleic acids and proteins can help diagnosis procedure [108]. Liang et al created an optimized double-filtration microfluidic system that separated and purified EVs (200 nm) from urine before quantifying them using a microchip ELISA (Figure 2) [109]. The amount of urinary EVs was markedly larger in patients with BC (*n* = 16) than in safe controls (*n* = 8) according to the observations. This optimized EV double-filtration system had a sensitivity of 81.3% and a specificity of 90%, according to receiver operating characteristic (ROC) analysis (16 BC patients and 8 healthy controls). As a result, this integrated technology has a lot of potential to be used in combination with urinary cytology and cystoscopy in clinics and at point-of-care (POC) settings to enhance clinical diagnosis of BC.

## 3. Applications of Nanomaterials in Treatment of BC

As mentioned earlier, BC is considered one of the most recurrent urogenital cancers in the world [110]. BC is an epithelial carcinoma that unusual cells in the epithelial lining multiply without any control. The most frequent histological class of BC is transitional cell carcinoma (TCC), also named urothelial cell carcinoma (UCC) [41]. Several methods are generally used to treatments of patients who suffered from BC, including chemotherapy, surgery of tumor, radiotherapy, immune therapy, stem cell transformation, and radical cystectomy. However, as we know, these methods will have many side effects such as restricted bioavailability, toxicity, non-selectivity, fast clearance, and limitation in metastasis. Therefore, researchers are studying to find newer treatment methods with fewer side effects and more therapeutic effects [111]. Generally, BC is a very heterogeneous and complex disease with various biological subtypes. So, it has many challenges in grading, classification. Approximately, 70% of patients who suffered from bladder UCC show a superficial carcinoma named *non-muscle-invasive BC (NMIBC)*, while the other patients (30%) progress a muscle-invasive carcinoma (MIBC) bearing the danger of the metastatic spread of the tumor [112]. The most common treatment for BC is surgery, but the findings have been shown that in approximately 80% of patients who underwent surgery, the tumor recurred after 5 years. Therefore, chemotherapy is still considered the main and important treatment in the inhibition of tumor recurrence and progression. Although chemotherapeutic agents, such as taxanes, cisplatin, gemcitabine, etc., facilitate prolong-term survival in many BC patients, the great recurrence rate of the tumor and serious adverse outcomes of therapeutic agents remain for BC therapy. Therefore, more effective and novel management is necessary to increase the quality and quantity of life of the BC patients [112].

Nowadays, nanotechnology has been dramatically helped us to diagnose and treat a wide variety of cancers like that BC. Recently, several nanoparticles (NP) such as polymeric NPs, lipid NPs, metallic NPs have been used to help BC therapy. Various forms of NPs can increase the solubility of drugs with weakly soluble, and multi-functional NPs have acceptable results against renal, bladder, prostate cancer. NPs are also utilized as a drug delivery system (DD) to increase effects and interactions between drugs and the urothelium. Furthermore, nanotechnology can also associate with other modern technologies to advance enhance effectively [113]. In this review, we intend to focus on nanotechnologies with potential and useful applications in BC treatment, also we will investigate several nanomaterials as nanocarriers to reduce side effects and enhance the effect of chemotherapy drugs to improve treatment of BC.

### 3.1. Some Important Therapeutic Agents for Treatment of BC

Recently, several efforts have been done to decrease the detrimental effects of therapeutic agents during the cancer treatment procedure to remove the side effects on the closed cells and tissues. Additionally, several methods have been found to enhance drug efficacy in the tumor, developing modern DDS and targeting systems [114]. Chemotherapeutic agents have classified into several groups involve cytotoxic and cytostatic agents which have shown acceptable effects alone or in combination with other cancer therapies. For example, Irinotecan (a semisynthetic derivative of the plant alkaloid) and doxorubicin (DOX) (the most popular anthracycline) are chemotherapeutic agents involve topoisomerase inhibitors, which used for many cancers. They have some important side effects such as neutropenia and cardiotoxicity, respectively. Other chemotherapy drugs such as oxaliplatin, carboplatin, cisplatin, melphalan, and cyclophosphamide are also considered alkylating agents. Their consumption is usually associated with the following side effects: gastrointestinal toxicity, cardiovascular toxicity, nephrotoxicity, and hematologic toxicity. The other chemotherapeutic drugs, such as vincristine, vinblastine, PTX, and docetaxel, are greatly suitable and useful against a broad range of cancers; however, these therapeutic agents are also showing particular restrictions such as their toxicity, side effects, expensive, etc., the major and common side effects of these anti-cancer agents are decreased blood cells, hair loss, and immunosuppression. The reason for these side effects is that they also target normal cells, which are rapidly dividing in the body. Consequently, nanomaterials are a promising drug delivery carrier, and used in therapeutic procedures designed to solve some of these side effects [114].

#### Role of Phytochemical Agents in BC Therapy

Phytochemicals are considered plant-derived materials that have a great effect on several diseases and can be useful for the inhibition of progressive diseases. Furthermore, phytochemicals have a considerable role as natural drugs in down regulating the multiple drug resistance (MDR) in different types of carcinoma particularly in BC [115]. In BC, curcumin (CUR) plays a major role, as it can prevent cell proliferation and decrease related reactions that lead to metastasis. These mechanisms are probably accompanied by down-regulating beta-catenin expression and reversing the epithelial-mesenchymal transition (EMT) process. Apigenin is another natural product capable to enhance amounts of reactive oxygen species (ROS), and can decrease the level of glutathione (GSH) in BC. On the other hand, Resveratrol capable represses cell proliferation and agitate apoptosis via signal transducer and activator of transcription 3 (STAT3) pathways and development of tumor in a xenograft model in BC. Therefore, phytochemical compounds could control the MDR process of UCC and increase the effect of anti-cancer drugs [115].

Phytochemicals have a wide volume of distribution, and caused accumulation in several organs. Moreover, the feasibility of the expansion of resistance through many pathways is the main challenge to the effective application of phytomedicines in cancer therapy. To control the challenge associated with conventional therapies, using nanomaterials are being performed [116].

In a recent study, Jung Cho and coworkers investigated several phytochemical agents such as CUR, capsaicin, quercetin, resveratrol, and their combination with gemcitabine. They have created a gemcitabine resistance urothelial cell carcinoma (UCC) cell line and then they used potential effective phytochemicals from phenolic groups mentioned above [115]. Western blot technique was utilized to find the expression of membranous ATP-binding cassette transporter isoform C2 (ABCC2) in *T24-GCB cells* and metabolic proteins, such as deoxycytidine kinase (DCK), thymidine kinase 1 (TK1), and TK2 in tumor cells. In this research, female BALB/c nude mice (NALC, ROC) were selected to confirm the effect of phytochemicals in combination with gemcitabine on BC therapy. The obtained results showed that Quercetin, CUR, and resveratrol have excellent effects with gemcitabine to *T24-GCB* cell lines. Resveratrol and CUR alone or with gemcitabine enhanced the level of ABCC2 but reduced cytoplasmic kinases simultaneously. On the contrary, Quercetin and capsaicin when used alone or with gemcitabine can diminish the expression of ABCC2 and DCK and TKs, in T24-GCB cell lines. The MDR of BC is strictly attributed to membranous ABCC2, cytoplasmic DCK, and TKs expression. Capsaicin has the powerful synergistic cytotoxic outcome of gemcitabine to *T24-GCB cells*. This combination system could be suggested as an adjunctive treatment for controlling MDR in BC [115].

In another study, combination therapy with curcumin and resveratrol in encapsulation in liposomes in male B6C3F1/J mice indicated a chemopreventive outcome. There was a considerable reduction in prostatic adenocarcinoma development following administration of the liposomal drug observed in in-vivo and in-vitro investigations through apoptosis activation and modulation of p-Akt, cyclin D1, mammalian target of rapamycin (mTOR), and androgen receptor (AR) [117].

One of the most important discussions about the operation of CUR in the preclinical and clinical treatment of BC is related to cancer cell growth and proliferation. After administration of CUR, cancer cells assemble at the mitotic (M) part of the cell cycle, representing a growth inhibitory outcome. It showed that CUR can inhibit BC cell proliferation [118]. Tian et al. have demonstrated that CUR actively controls tumor proliferation by suppressing the PI3K/AKT/mTOR signaling pathway in a rat BC model. on the other hand, CUR can decrease the upregulation of Insulin-like growth factor 2 (IGF2), known as protein hormone, and the phosphorylation position of its ligand IGF1-receptor (IGF1-R) and insulin receptor substrate 1 (IRS-1) which can transmit signals to PI3K. With these results, the researchers accept that CUR suppresses the function of the IGF1-R/IRS-1 pathway [119].

The other research investigated the efficacy of epigallocatechin gallate (EGCG) EGCG-GNPs in a mouse model of BC and observed the inhibition of tumor cell development by apoptosis [117].

In another study, CUR was conjugated with cyclodextrin (CDC), as a type of nano-material, to improve the solubility of CUR. Both human urothelial carcinoma cell lines and the AY-27 rat cell line were treated to different levels of CDC in vitro. For the in the vivo study, the AY-27 orthotopic BC F344 rat model was utilized. Rats were treated with consecutive intravesical instillations of CDC, *Bacillus Calmette Guérin* (BCG), the combination of both of them, and NaCl as control. CDC indicated an antiproliferative effect on rat and human urothelial carcinoma cell lines in vitro experiment [120].

In another research, Park et al. investigated the apoptotic effects of the CUR accompanied with cisplatin as co-treatment in *T24 BC* cell line. Moreover, they carried out an in vivo study on nude mice bearing 253J-Bv cell xenografts. The finding showed that approximately one month after cisplatin–CUR combination therapy, a significant reduction in tumor size, whereas no response was observed when CUR or cisplatin was used alone [121]. Remarkably, CUR does not cause symptoms of toxicity in animals, which indicates its safety. Although many studies are needed to find out exactly how CUR works in a cisplatin-based treatment, concomitant utilization may provide an innovative method to manage human BC [118]. Recently, Miyata and coworkers investigated treatment methods by using fucoidan in BC. Fucoidan is a sulfated polysaccharide derived from marine brown algae known as a multi-functional and non-toxic substance and has anti-cancer functions in different types of malignancies. Furthermore, several studies have demonstrated the positive outcomes of fucoidan versus cancer-related dyspepsia and chemotherapeutic drug-induced adverse events [111]. Han et al. have shown the correlation between fucoidan and apoptosis in vitro investigations. They utilized DAPI staining (4, 6-diamidino-2-phenylindole) and flow cytometry in the human BC 5637cell line [122]. They have treated BC cells with 0, 10, 25, or 100 µg/mL of fucoidan for 24 hours. Subsequently, nuclear multi-fragmentation and chromatin condensation have been seen which depend on the concentration. Furthermore, results obtained from flow cytometry indicated that the percentage of cells with sub-G1 DNA content, as a parameter of apoptosis, was enhanced in a concentration-dependent manner. So, the pro-apoptotic consequence of fucoidan was recognized in BC cells using different methods [122].

### 3.2. Applications of NPs to the Treatment of BC

Recently, nanotechnology has been widely advanced for the treatment and diagnosis of different carcinoma such as both NMIBC and MIBC [123]. Investigation into NPs for various types of cancer has shown a great number of applications; however, many of them remain in the preclinical state [78,124,125,126,127,128,129,130,131]. Several NPs, such as metal NPs and polymeric NPs, can be suitable to early diagnostics of cancer diagnostics and as a potential nanocarrier for loading chemotherapeutic drugs [123].

Many of the therapeutic agents can be encapsulated or conjugated to NPs and can be targeted both actively and passively to the tumor location. As we know, tumor tissue has an unusual and leaky vasculature, which causes the NPs to accumulate simply. This process is also known as the enhanced permeation and retention (EPR) effect, which was widely utilized. However, passive targeting also has limitations; there is the possibility of inappropriate targeting and dispersive effect of the drug on tumor cells. Furthermore, the EPR effect is very dependent on the intrinsic tumor biology, and not all tumor environments show the EPR effect. Therefore, in such cases, active targeting can be used. In active targeting, several molecules such as peptides, vitamin, and antibodies (Ab) conjugate to the NP surface. After that, these molecules attach to their receptor sites of tumor cells, and release the drug during the endocytosis process [132].

Several of NPs systems are currently investigated in the preclinical and clinical phases. In this work, we investigated data showing the usage of NPs in BC therapy, which can diminish side effects of therapeutic agents and recurrence grades of tumor. We also summarize the various types of NPs that have been applied for BC therapy [41]. In the other hand, staging of BC is categorized according to the location and proliferation of tumors (Figure 3), Ta (low-risk tumors): non-invasive papillary carcinoma; T1: The tumor has developed from urothelial section into connective matrix; T2: The tumor has invaded into the muscle level; T3 showed the tumor has penetrated via the muscle level and into the fatty tissue; T4: The tumor has developed outside the fatty tissue and into near organs like prostate or vagina. In each of these stages, NPs such as liposome, polymeric micelle, metal NPs, natural NPs like exosome can play an effective role to prevent the spread of BC.

In Table 1, we describe the existing state of NPs in the treatment of BC [123].

#### 3.2.1. The Applications of Various Metal NPs to Treatment of BC

Metal NPs, such as silver and gold, play beneficial role in cancer therapy. Metal NPs potentially have acted simultaneously in diagnostic and treatment, and permit targeted drug release. Furthermore, functionalized metal NPs with targeting ligands can be more useful and effective strategy for removing tumor [141].

##### Gold NPs (GNP)

GNPs have been employed in order to diagnose and treat a wide range of tumors for a long time. Its attributes, such as high surface to volume ratio, stability, and easy synthesis, as well as its non-toxic nature, have led to its use as a nanocarrier of many drugs, and, also, authorizing the accumulation of therapeutic agents at the tumor environment (Figure 4) [132].

GNPs are playing a very important role as drug carriers because of their low toxicity, compatibility with patient’s cell, surface plasmon resonance (SPR), optical, and tunable properties (Figure 4). They can be prepared in a wide-ranging core between 1 to 150 nm diameters, which lead to it easier to adjust their dispersion. Moreover, having a negative charge of GNPs has caused provides them simply modifiable. This means that they can be designed simply by the decoration of various biomolecules, drugs, targeting ligands, and even genes [132].

In a recent study, Botteon et al. showed that biosynthesis of GNPs utilizing Brazilian Red Propolis (BRP) plant extract is indicated as an easy and low-cost method. BRP is a useful material obtained from bees that have exhibited several significant properties such as anti-tumor and anti-oxidant. Botteon et al. reported the bio-synthesis of GNPs utilizing BRP extract (GNP_extract_) and its component such as liquiritigenin, formononetin, vestitol, guttiferone E, with several fractions of GNP _hexane_, GNP _dichloromethane_, GNP _ethyl acetate_ obtained. They also assessed their physical and biochemical properties. They found that the most important features were their potential function against tumor cells. GNPs indicated dose dependent cytotoxic function both in T24 and PC-3 cell lines. GNP _dichloromethane_ and GNP extract showed cytotoxic effect in vitro. As a result, they show that the BRP hydroethanolic extract and its fractions have a great potential to produce GNPs with diameter range between 8 and 15 nm. GNP extract demonstrated antifungal properties with high cytotoxicity and negligible concentrations in BC and prostate cancer cells. Dichloromethane and hexane fractions obtained from GNPs exhibited great antibacterial and antifungal functions and cytotoxicity in *T24 and PC-3* cells studied [142].

Xing et al. in recent study investigated novel chemotherapeutic GNPs were equipped to treat BC [143]. GNPs were produced by *Citrus aurantifulia* seed extract as the capping factor. Characterization of GNPs was carried out with several important device such as Fourier-transform infrared (FT-IR), transmission electron microscopy (TEM), UV-Vis, energy-dispersive X-ray spectroscopy (EDS), and field emission scanning electron microscopy (FE-SEM). In UV–Vis, the exact peak in the wavelength of 522 nm showed the formation of GNPs. 3-(4,5-dimethylthiazol-2-yl)-2,5-diphenyl-2H-tetrazolium bromide (MTT) assay was done on BC in several cell lines including (HT-1376, Grade 3, carcinoma), TCCSUP (Grade IV, transitional cell carcinoma), SCaBER (squamous cell carcinoma), and UM-UC-3 (transitional cell carcinoma). GNPs showed very low cell viability and anti-BC properties dose against HT-1376, TCCSUP, SCaBER, and UM-UC-3 cell lines. The great result of the anti-BC properties of GNPs was obtained in the case of SCaBER cell line. Furthermore, these GNPs were appropriate suppressor of the cholinesterase and α-glycosidase enzymes. After confirming these outcomes in the clinical trial investigations, GNPs can be utilized as anti-oxidant, anti-diabetic, anti-cholinergics, and anti-BC supplements in human [143].

Jazayeri and coworkers demonstrated a fantastic method for diagnosing of BC by using of several biomarkers, containing survivin. Survivin has been known as a protein with the main function in cancer development, by handling the level of cell apoptosis. Obtained information demonstrates that there is not present survivin in ordinary tissues, while the amount of survivin expression enhance in cancer cells. The main purpose of this project was to control the reactive Ab to survivin protein as a biological factor to control the BC grade with ELISA procedure and applying GNPs conjugated with survivin Ab. As result, the concentrations of surviving in serum and urinary were considerably enhanced in high grades of the BC patients than the normal individuals. Additionally, using conjugated GNPs, survivin protein was identified in the urine samples of patients at all grades. Their results indicated that utilizing the ELISA process, the enhanced concentration of survivin could be helped us to distinguish in high grades of BC, but using anti-survivin antibody-conjugated GNPs, BC can be identified in early stages [38].

##### Iron NPs (FeNP)

Iron oxide (FeNPs) is considered as a biocompatible nanomaterial that provides super paramagnetic DDS, which can be utilized to the targeted tumor site by the external functional magnetic field [112]. Zakaria et al. developed nanoporous FeNPs and utilized them to the intra-cellular DDS of BC cells. It is important that the DDs efficiency was improved by utilizing magnetic conduction. Additionally, treatment with helping magnetic targeting can considerably enhance the level of the therapeutic agent in the cell. Employing FeNPs, therapeutic doses and adverse effects of therapeutic agents can be decreased. Additionally, FeNPs with fantastic properties can be used in optical imaging. In many types of research performed on animal models suffering from BC, the drug distribution could be a long time remained, and the tumor cells could be imaged through MRI [112].

In another study, a group of researchers used chitosan with covered super paramagnetic iron oxide NPs (CSSPION). They were prepared and utilized a nano-vehicle for loading of chemotherapy drug 5-Fluorouracil (5-FU), CS-5-FU-SPION, via a reverse micro emulsion method. In the last step of the preparation procedure, the nanostructure complex was designed with folic acid (FA-CS-5-FU-SPION) to targeted therapy. This nanosystem was performed on the *T24 BC* cell line. The information showed that the FA-CS-5-FU-SPION has spherical stricter and obtained result from dynamic light scattering (DLS) showed an average diameter size 79 ± 13 nm. Furthermore, the notable drug loading efficiency obtained approximately ~73%. The results indicated that FA-CS-5-FU-SPION displayed anti-tumor characteristics on cancer cell without any adverse effect on normal cells. Moreover, it became confirmed that the fluorescein isothiocyanate (FITC) labeled FA-CS-5-FU-SPION, has successfully passed into tumor microenvironment and trigger cell death and apoptosis [134].

##### Silver NPs (AgNPs)

Today, AgNP is used to treat a variety of cancer cells, mainly due to its antineoplastic properties, which enable it to simulate cancer cell death in the same way traditional chemotherapy does. The functions of Ag NPs anti-tumor activity probably is correlated with the releasing of metallic silver (Ag^0^) and silver cations (Ag^+^), which can both cause oxidative stress, mitochondrial and DNA damage, phospholipid bilayer membrane destruction, and genotoxicity, outcoming in cell death by necrosis or apoptosis [144].

In a recent study, scientists studied the anti-tumoral property of NPs obtained via Ag biomass of the Fusarium oxysporum, a type of fungi, then they assessed in 5637 cell line of human BC. The result showed the cytotoxicity, molecular function of cell death, and prevention of cell migration and proliferation. This study indicated the function of AgNP to the NMIBC induced in a group of female mice (the C57BL/6JUnib model) by AgNP intravesical administration. Their results demonstrated that, in BC cells, AgNP caused DNA double-strand breaks and lead to apoptosis, which reduces migration and cell proliferation. So, these results confirmed that AgNPs could be an effective tool for the treatment of BC [144].

In another study, Zhao et al designed poly-dopamine-coated branched Au–Ag NPs (Au–Ag @ PDA) to inhibited cancer cell proliferation. They studied the toxicity of the Au–Ag @ PDA NPs against T24 human BC cells. Additionally, it is noteworthy that molecular mechanisms of photothermal therapy can stimulate T24 cell apoptosis [7]. In this procedure, T24 cell lines were treated with various doses of complex Au–Ag @ PDA NPs and then applied by 808 nm laser diode. T24 cell lines were treated with various doses of complex Au–Ag @ PDA NPs and then employed by 808 nm laser diode. Finally, several items, such as cell growth, cell cycle, autophagy, and apoptosis, were investigated [7]. In addition, they assessed the results of treatment on mitochondrial membrane and ROS generation to support the main mechanisms of inhibition. As a result, they investigated the inhibitory functions of T24 Au–Ag @ PDA NPs on tumor utilizing a xenograft mouse model [7]. They established that Au–Ag @ PDA NPs, using appropriate laser irradiation, reduced the growth of T24 cell lines and can alter the cell cycle by increasing the proportion of cells in the S stage. Furthermore, apoptosis cell by motivating the mitochondria-mediated intrinsic pathway increased, and a strong autophagy reaction in T24 cells activated. The important report about this work is, Au–Ag @ PDA NPs can suppress the expression of phosphorylated AKT and extracellular signal-regulated kinase (ERK) signaling pathway and increased the level of ROS that function upstream of autophagy and apoptosis. Additionally, in vivo study have been showed that Au–Ag @ PDA NP with photo-thermolysis also considerably suppressed tumor growth [7].

#### 3.2.2. Applications of Polymeric Micelles to Treatment of BC

Recently, polymeric micelles, due to their spherical structure, which comprise amphiphilic copolymers, have gained great attention in nanotechnology [145]. Figure 5 shows a general schematic, from delivery of therapeutic agent loaded within micelle as nanocarrier in order to release the drug to tumor microenvironment. As we know, amphiphilic copolymer can include of di or tri block copolymer and graft copolymers. The self-assemble of copolymer contain of hydrophobic core and hydrophilic shell can form micelle with the size range of 10 to 100 nm [146]. The structures of polymeric micelles have potent stability and they can increase solubility of hydrophobic drugs. In addition, the nano sizes of polymers prevent clearance of micelles, so it is useful for prolonging the blood circulation of drugs. Additionally, the outer shell of micelles can control biodistribution of polymeric micelles. For this purpose, the corona or outer shell of the micelles can be modified by using of hydrophilic polymer like poly ethylene glycol (PEG) [147]. These hydrophilic polymers provide a ‘stealth’ effect to the nano carrier. Therefore, it can increase circulation in the blood by avoiding detection, and gaining uptake through the mononuclear phagocytic system (MPS) or reticuloendothelial system (RES). The critical micelles concentration (CMC) is an important key parameter for the formation of micelles at critical concentration. The CMC of polymeric micelles is (1000 times) lower than CMC of surfactants. As a result, the lowest CMC is preferable for drug retention in polymeric micelles following intravenous (IV) administration at high dilution [148].

Many researchers applied polymeric micelles in order to treatment of BC, for instance, Zhou and et al. designed a targeted DDS with potentials for intravesical instilled chemotherapic agents in order to treatment of superficial BC [135]. They used amphiphilic di-block copolymer poly (ε-caprolactone)-b-poly (ethylene oxide) (PCL-b-PEO), which polyethylene oxide (PEO) is considered as a hydrophilic block and poly-caprolactone (PCL) as a hydrophobic block. First, cyclic arginine-glycine-aspartic acid-D-phenylalanine-lysine C (RGDfK) peptide and florescent label was conjugated with the copolymer and formerly accumulated into micelles. C (RGDfK) peptide showed high affinity to αvβ3 integrin, which overexpress on the surface of T24 cell lines, when DOX was loaded into the system. As a result of the confocal analysis, it was discovered that encapsulating DOX through the c (RGDfK) peptide would aid micelles in targeting tumor cells with strong anti-proliferation of tumor bladder cells [135].

In another similar study, another targeted drug delivery system was investigated in order to intravesical instilled anti-cancer agent. In this study, both folic acid (FA) and fluorescein isothiocyannate (FITC) was decorated to PCL-b-PEO-NH_2_ copolymer structure. The cellular uptake mechanism for *T-24* cells applying confocal demonstrated that PCL-b-PEO-FA has more uptake, due to overexpression of FA receptor on the surface of carcinoma cells. In vitro, the adjusted with FA has much lower cell viability with DOX loaded in micelles than without targeted micelles, according to the cell cytotoxicity experiment. Additionally, this structure can reduce adverse effect of DOX via receptor mediated endocytosis mechanism [135].

Zhong and et al. designed a suitable method in order to development a polymeric micelle structure by using DOX as chemotherapy drug which is covalently conjugated to tri-methyl-chitosan (TMC) with beclin-1 siRNA (Si-Beclin1/DOX-TMC) [149]. The si-beclin1/DOX-TMC micelle displayed more cytotoxic effect to both drug-sensitive BIU-87 and drug-resistant BIU-87/ADR cell lines. This formulation showed strong capacity for autophagy in BC cells. In fact, DOX may cause inducing protecting autophagy in this type of cell line. It is noteworthy that, the equipped si-beclin1/DOX-TMC micelle prevents the beclin-1 protein expression to prevent protective autophagy of BIU-87/ADR cells, but simultaneously inducing apoptosis mechanism. In fact, they can demonstrate a new strategy to overcome chemotherapy resistance by using nanomaterial. Since that, regulating the apoptosis and autophagy are combined by suitable design and preparation of multi-functional biocompatible nanocarriers of siRNA and drug co-delivery. So, codelivery of DOX and becline-1 SiRNA described synergistic effect to suppress drug resistant BC. In vivo investigation was carried out in BIU-87/ADR xenograft BC models and si-beclin1/DOX-TMC nanomicelles showing significantly suppressed tumor growth [149].

Pan et al developed nanosized micelles loaded with PTX and investigated anti-tumor effect against BC [136]. Structure of micelle-formed telodendrimers was synthesized via conjugation of 8 units of cholic acid at one end of PEG and a BC-specific targeting peptide named PLZ4 (cancer specific ligand) at the other end. It should be noted that, cysteine (Cys) was presented between the cholic acid and PEG when prepared disulfide cross-linked PLZ4 nanomicelles (DC-PNM). Experimental results and anti-tumor effect were confirmed in mice with immune system inefficiency, carrying patient-derived BC xenografts (PDX). After IV administration, DC-PNM definitely targeted the BC PDXs. Furthermore, DC-PNM loaded with PTX can remove cisplatin resistance, and increase the median survival from free PTX compared to mice carrying PDXs. As result, DC-PNM exhibited stable in the sodium dodecyl sulfate (SDS) solution and precisely targeted the BC xenografts. Thus, the system enhanced the efficacy of PTX as an anti-cancer drug [136].

In another study, the combination of micelles and PLZ4 was evaluated for studding of imaging and treatment in BC cases. PTX, as an anti-cancer, were loaded to decorated micelles. The targeting micelles accumulated in cytoplasm of 5637 human cells and T24 cell line. The result showed enhancement of drug delivery in target site. Additionally, reduce PTX toxicity and increase overall survival [150].

#### 3.2.3. Applications of Polymeric NPs to Treatment of BC

Polymeric based NP system are the most used among the NPs that can be prepared from several polymers, including natural or synthetic materials composed of macromolecules such as poly (lactide-coglycolide) (PLGA), poly(lactic acid) (PLA), poly(e-caprolactone)(PCL), poly(alkyl cyanoacrylates), and chitosan. Due to physicochemical structures of polymeric NPs, we intended to investigate solid NPs, polymeric micelles, dendrimer system [151]. Advantages of polymeric NPs have attracted interest over the past years. For example: the suitable size around 100–300 nm, or even less than 100nm; are able to control release; less expensive; protect the drug against environmental reactions [152]. However, polymeric NPs are toxic to patients, for example, PLA have poor toughness, degradation speed slow or chitosan have poor strength, low water-solubility [153]. So, it is essential to improve their biocompatibility and reduce toxicity in order to biomedical usage. As we know, after resection of the tumor in order to treatment of patients who suffer from BC, chemotherapy is considered as a standard way for preventing the development of the disease in the patients. However, the high rate of excretion of chemotherapeutic drugs is a major challenge in their effectiveness. It is noteworthy that polymers with mucoadhesive features can improve the effectiveness and penetrability of chemotherapeutic drugs. Therefore, it could be help to increase retention time of drug delivery in the post-surgery treatment of BC patients. In a recent research, scientists designed a promising drug delivery system by chitosan as a positively charged polymeric NP system in order to increase the adhesion and permeability of prodrug gambogic acid within the bladder wall. They used a reduction sensitive carrier to transport the reactive oxygen species (ROS) activated prodrug of gambogic acid in order to treatment of orthotopic BC. Furthermore, they examined the various level of glutathione (GSH) and ROS between normal and cancer cells, the dual responsive NP carrier can selectively deliver the drug into BC cells. As result, the system can significantly prevent the growth of BC cells in an orthotopic superficial BC model without leading to destruction to normal cells [137].

In another study, Li et al. designed fluorinated chitosan nanosystem in order to increase transmucosal delivery of sonosensitizer-attached catalase for sonodynamic (SDT) BC therapy post-intravesical instillation. The SDT nano-platform which a transmucosal O2-self production is designed in order to have more efficient SDT against BC [154]. In the study, chitosan decorated with flour (FCS) is made as a more effective safe transmucosal delivery nanocarrier to accumulate with meso-tetra (4-carboxyphenyl) porphine-bonded catalase (CAT-TCPP). The engineered CAT-TCPP/FCS NPs after intravesical instillation into the bladder showed very good intratumoral, and also transmucosal penetration. Furthermore, it could efficiently produce hypoxia in tumor microenvironment by the catalase-catalyzed O2 generation from cancer cells endogenous H_2_O_2_ to further develop the therapeutic efficacy of SDT to ablate orthotopic BC cells under ultrasound.

In another work, Zhu et al prepared DOX and IR780@PEG-PCL-SS NPs in order to create a chemo photothermal therapy. In photothermal therapy, light absorption makes photosensitizers generate heat. It can cause all of target cells to undergo necrosis and apoptosis. Thus, this method can promote therapeutic efficacy. In this study, the formulation of NP which contain DOX and IR780 are sensitive to presence of GSH in micro-environmental of BC cells and near-infrared laser irradiation. Therefore, the NP can release drugs under near-infrared laser irradiation (IR) in the presence of GSH, which enable the controlled the release of drugs. The result of the in vivo study demonstrated that significant efficacy and suitable size of tumor obtained after 21 day of treatment [140]. In another research, poly (ethylene glycol)-b-(poly(ethylene diamine l-glutamate)-g-poly(ε-benzyoxycarbonyl-l-lysine)-r-poly(l-lysine)) (PEG-b-(PELG-g-(PZLL-r-PLL)) were prepared by Hu et al. lumbrokinase as an enzyme which has a half life that is too short to be used in combination with other drugs. Due to its negative charge, it combined with PLL brush block via electrostatic interaction. PTX with hydrophobicity property is able to combine PZLL through hydrophobic interaction, thereby enhancement of the half-life and bioavailability of the drugs could be developed via the sustained release and enhancement of tumor site enrichment through passive targeting. They found that (PEG-b-(PELG-g-(PZLL-r-PLL)) polymers prolonged the half-life and bioactivity of LK and PTX in Sprague-Dawley rats [155]. Tao et al. developed increasing delivery efficiency with utilizing different degree of cholesterol substituted pullulan polymer (CHPS). The best results were provided when the degree of cholesterol was high with the smallest size and efficient loading. The result demonstrated that CHp-3 with largest size had strongest ability to prevent of cells migration. Additionally, all three type of NP which containing drugs could inhibit MB49 cells [156].

#### 3.2.4. Applications of Liposome to Treatment of BC

Liposomes are synthetic phospholipid nanovesicles with a bilayer membrane shape, first developed by Alec Bngham in 1961 [138]. Liposomes have been used in carrying various biological and biochemical molecular such as drug, nucleotides, protein, etc. The binding of liposomes to human BC cell lines affects the usage of liposomes in BC therapy. The utilization of liposomes show great advantages as nanovehicles, such as being capable of solubilizing both hydrophilic and hydrophobic molecules in an aqueous core and phospholipid bilayer, respectively; biocompatibility because of their origin from cholesterol and phospholipids; good stability and capability to transport drugs by suitable selection of preparation procedure [42]. Nogawa et al. reported that polo-like kinase 1 (PLK-1) is related to grades and survival rate of BC patients. They carried out, intravesically, instilling a complex of PLK-1 siRNA/liposomes into the orthotopic tumor bladder of the mouse. They successfully exhibited transfecting PLK-1 siRNA into tumoric cells and diminished PLK-1 expression and suppression of cancer proliferation in this mouse model [138]. Intravesical administration of anti-proliferative material seems one of the common therapeutic methods for treatment of BC. Liposomes encapsulated with therapeutic agents have been confirmed to enhance the efficacy of intravesical therapy.

Hsu et al. reported that recombinant human interferon alpha (IFN-α) loaded in liposomes inhibits growth of BC cell line 253J more than free IFN-α alone or liposomes with saline. Furthermore, the same investigation also established that a subline of 253J, which is resistant to free IFN-α, became responsive to IFN-α loaded in liposomes [138]. Intravesical instillation of live bacillus Calmette Guérin (BCG) bacteria utilized as an ideal treatment method for high grade NMIBC. However, serious adverse effects were reported, including containing BCG infections, sepsis, and even death. Repackaging parts of the BCG bacteria has been carried out in order to substitute for live bacteria with the promise to achieve safe and suitable efficacy. For instance, the cell wall skeleton (CWS) of BCG was investigated for clinical application. However, it was seems to be more hard to dissolve in water and deliver to the site of the target. By encapsulating BCGCWS within liposomes, whose size was 166 nm, Nakamura et al. were able to report suitable efficacy in rat models against the development of BC [123].

The *mycobacterium bacillus calmette Guérin* cell wall skeleton (BCG-CWS) was considered the important immune active part of BCG. So, it is a great suggestion as a non-infectious immunotherapeutic drug, and also as a substitute to live BCG to utilize against urothelial cancer. However, its usage as anti-cancer therapy is restricted due to the fact that BCG-CWS can aggregate in all aqueous and non-aqueous solvents. To increase the internalization of BCG-CWS into BC cells without aggregation, Whang and coworkers in 2020 designed an interesting nanosystem consist of liposome which loaded nanosized BCG-CWS (180 nm) for intravesical instillation in BC. This nanosystem displayed an anti-tumoral effect in an orthotopic BC mouse model, and the BCG-CWS NPs can be utilized as a non-toxic substitute for live BCG with improved stability, and size compatibility. They used an intravesical way by applying a catheter in the orthotopic BC mouse model in order to intravesical delivery of BCG-CWS mouse to selectively target tumor cells. In vitro results indicated that BCG-CWS nanosize, encapsulated with conventional liposomes (CWS-Nano-CL), was more effective at inhibiting BC cell growth compared to nonencapsulated BCG-CWS. Treatment with CWS-Nano-CL prompted the inhibition of the mammalian target of rapamycin (mTOR) pathway and also the activation of AMP-activated protein kinase (AMPK) phosphorylation, cause to apoptosis, both in vitro (5637 cell lines) and in vivo. So, they suggested that the intravesical instillation of CWS-Nano-CL can help BCG-CW cellular endocytosis and supply a smart drug-delivery system as a therapeutic method for BCG-mediated BC treatment [133].

Recently scientists designed a drug delivery system consist of maleimide-modified PEGylated (Mal-PEG) liposomes were known as mucoadhesive nanocarrier for intravesical therapy of BC. Mal-PEG liposomes loaded fluorescein sodium (FS) displayed more penetration and retention time on bladder mucosal compared to without liposomes. In another work a group of scientists have engineered a platform by utilizing fluidizing liposomes incorporated into gellan hydrogel which provide an in situ-gelling liposome-in-gel (LP-Gel) system. LP-Gel employs urine to undergo ion-triggered gelation to construct a cross-linked gellan matrix. The platform mimics the bladder mucosa, so resulting in suitable interaction and adhesion to the bladder wall. After LP-Gel platform is instilled into the bladder of rat, the ion-triggered gelation binds to the urothelium, after that, the fluidizing liposomes penetrate via the urothelial barrier and the drug localization in tumor lesions were prolonged. Furthermore, instillation of paclitaxel-loaded LP-Gel showed protracted drug localization in the bladder almost seven days, offering potential usage in clinical practice [157].

In a recent study, Valenberg et al used 1.2-dipalmitoylsn-glycero-3-phosphodiglycero (DPPG2) based thermosensitive liposomes (TSL) with loaded DOX combined with hyperthermia (HT) caused greater amounts of DOX in the bladder. In this work, they used 21 pigs in which, after anesthesia, they were placed in a urinary catheter equipped with a radio frequency antenna for HT (1 hours), and then administrated different doses of DPPG2-TSL-DOX and free DOX with or without HT. After that, the pigs were immediately sacrificed. HPLC was utilized to measure DOX concentrations in the bladder, vital organs and serum, and fluorescence microscopy was used to estimates the distribution of DOX in the bladder wall. They concluded that this method compared to conventional DOX, increases the level of DOX in the bladder wall while diminishing DOX in the major organs. DPPG2-TSL-DOX accompanied with HT provided minor DOX accumulation in the heart and kidney of pigs, compared to IV free DOX [139].

In another similar study, Mikhail et al used lyso-thermosensitive liposomal doxorubicin (LTLD) in combination with local mild HT, in order to targeted drug delivery to the bladder and potential BC therapy. This study also performed on porcine with the following three categories: (I) IV administration of LTLD with HT, (II) IV free DOX with HT, and (III) IV LTLD without HT. Formulations of the drug were delivered through IV infusion which prolonged 30 min, simultaneous with 1-h bladder irrigation, 45 °C water for HT category, 37 °C for non-HT category, then immediately bladder resection. DOX levels were investigated in consecutive parts parallel to the bladder lumen by liquid chromatography following drug extraction. As an outcome, they established that DOX accumulation and distribution within the bladder was obtained at higher amounts than with free IV DOX by mild bladder hyperthermia accompanied by systemic delivery of LTLD [158].

#### 3.2.5. Applications of Exosome as Natural NP to Treatment of BC

Exosomes are tiny endosome-derived membrane nanovesicles (approximately 30–100 nm) secreted from various types of host cells showing different biological function. Exosomes have a noteworthy role in cellular communications as a vehicle of nucleic acids and proteins. They have been emerged as novel mediators of tumor progression over the recent decade [159]. Exosomes have multiple and important biological roles, such as immune response regulation, presentation of antigens to immune cells, intercellular relation via transfer of proteins, nucleic acid such as miRNA, mRNA. Exosomes are a major source of tumor biomarkers in liquid biopsy samples. Exosomes can separate from the urine of BC patients with MIBC induced epithelial to mesenchymal transition in urothelial cells. A novel observation into the function of exosomes in the transition of BC into muscle-invasive cancer was carried out. Therefore, exosome investigation in advanced BC could be considered a modern system for predicting progression and innovating targeted therapy [160]. Exosomes create by the inner budding of intra-cellular endosomes, after that altered into multi-vesicular bodies (MVB) that belong to the endosomal pathway and can fuse with the cellular membrane and secrete their payload into the extracellular environment [161]. Generally, exosomes existent in most biological fluids and show important functions in cancer. So, investigations amount of exosomes in various steps of BC could be useful for diagnosis and timely therapy of BC [159].

Newly, scientists confirm that exosomes are a potential source of tumor biomarkers in liquid biopsies, such as blood and urine samples due to containing RNAs, DNAs, and proteins. In particular, exosomal miRNAs (exomiRs) play an important role as biomarkers for tumor growth and progression (Figure 6) [162].

Body fluid biopsy is one of the most promising methods in cancer research because it is a simple, low-cost, and non-invasive procedure. So that, liquid biopsy specimens can show us useful and complete information about the genetics of cancer patients.

Elsharkawi et al. reported that tumor derived exosomes (TDE) could be a useful and suitable biological device for early identification of BC. They were measured TDEs concentration in urine and serum samples of seventy BC patients from Ta to T3 stages, and 12 healthy control people using ELISA technique. As results show, exosomes concentrations in BC patients were enhanced compared to healthy people in serum and urine samples at different phases of the disease. Moreover, they found that serum was more particular sample for discovery of exosomes in BC [159].

Hiltbrunner et al. reported that exosomes with pro-carcinogenic properties could be identified in urine from histologically down-staged BC patients. They demonstrated that urinary exosomes from the bladder, even when no macroscopic tumor observe. Next, a combination of primary transurethral resection (TUR-B) and neoadjuvant cisplatin-based combination chemotherapy (NAC), vary from exosomes found in urine from the upper tract. These exosomes exhibited a malignant metabolic manner, which probably could enhance metastasis and recurrence. Hiltbrunner et al. reported that the bladder’s performance as a site for exosomes capable to disseminate to far sites in lymph nodes and other distant organs, where they help dispersion via metabolic communicating [163]. In fact, they found that why a large number of MIBC patients recurrence even after NAC and radical cystectomy (RC). They showed that exosomes hold a malignant memory phenotype in the bladder even after (TUR-B) and NAC, confirming the significance of radical cystectomy over minor surgery to remove the original tumor-promoting exosomes. However, further studies on the metabolic index of urine and bladder muscle are needed to prove this fact [163].

Goulet et al. demonstrated that extracellular vesicles (EV) including microvesicles, apoptotic bodies and exosomes derived from BC (BCEV) patients can enhance “transformation” of normal fibroblasts into cancer-associated fibroblasts (CAF). They isolated extracellular vesicles from several cell lines such as *T24, RT4,* and *SW1710 BC* and utilized them to treat normal fibroblasts separated from human bladder biopsies. The outcome showed that recipient fibroblasts obtained CAF composition with enhanced proliferation and migration ability, as well as raised expression of CAF markers smooth muscle actin (SMA), fibroblast motivation protein (FAP), and Galectin [164].

In recent years, researchers have reported exosomes can be applied as nanocarriers, which can be utilized in the treatment of many tumors by releasing their cargoes into the targeted site. exosomes as nanocarriers to load anti-tumor therapeutic agents or siRNAs into exosomes. The advantages of this method including improved the efficacy of drugs, enhancing the bioavailability of drugs; being non-toxic or low-toxic, etc. For example, Wang et al established a dual-functional exosome based superparamagnetic NP system employing exosomes as a targeted drug carrier for the treatment of cancer cells [165].

In another study, Cai et al reported that the effect of exosomal miR-133b on the proliferation of BC and its molecular pathway. First, they have investigated the expression of miR-133b in BC and neighboring healthy tissues, as well as in body fluid exosomes of BC patients and healthy controls. Then, the entry of exosomes in cells was confirmed via fluorescence localization. Furthermore, cytotoxicity and apoptosis were carried out in BC cells transfected with mimics and incubated with exosomes. After that, the function of exosomal miR-133b was investigated in nude mice trans-plant tumors. Moreover, the target gene of miR-133b was estimated via bio-informatics technique. The amount of miR-133b was considerably reduced in BC tissues and in exosomes from the fluids of patients, correlating with weak general survival in The Cancer Genome Atlas (TCGA) database. Exosomal miR-133b can be obtained using post-transfection BC cells by mimicking miR-133b. MiR-133b expression was enhanced after incubation with external miR-133b, leading to inhibition of viability and increased apoptosis in BC cells. Exosomal miR-133b can suppress tumor growth in vivo. In addition, we found that exosomal miR-133b might be involved in inhibiting BC proliferation by rearranging the dual-specific protein phosphatase 1 (DUSP1). These findings could hold new hope for BC treatment directions [166]. In conclusion, their results provide strong evidence that exosomal miR-133b acts as a tumor inhibitory agent targeting DUSP1 in BC proliferation. In addition, exogenous miR-133b can be ingested by BC cells, reducing the malignant phenotype of BC cells. Exosomal miR -133b or other miRNAs better distinguished the disease and make specific advances in the treatment of BC [166].

## 4. Conclusion, Challenges, and Future Prospective

Many attempts have been made in recent decades to fight cancer by combining novel technology and traditional approaches. In this article, the applications of nanotechnology were discussed. Nanotechnology has demonstrated extensive applications both in the diagnosis and in treatment of variety of cancers, including improving the selectivity and sensitivity for the treatment of BC. NPs have been able to overcoming the limitations of current medical approaches in many research and clinical trials. For MRI and CT imaging, various metallic NPs are applied as contrast agents, increasing the likelihood of detecting early BC. GNPs’ excellent photothermal properties have contributed to their widespread use in imaging. However, several concerns remain, such as the protection of these NPs, when administrated in the human body. Additional research is required to fully understand the specifics of nanotechnology-based methods, and to develop these NPs as cancer diagnostic tools. On the other hand, while NPs have the potential to treat BC, there are still a number of hurdles to overcome before, they can be successfully translated into clinics, including research into the relationship between NPs and BC cells. The metabolism, biodistribution, and clearance of NPs, as well as the composition of their target cells, are also unknown.

## Figures and Tables

**Figure 1 cancers-13-02214-f001:**
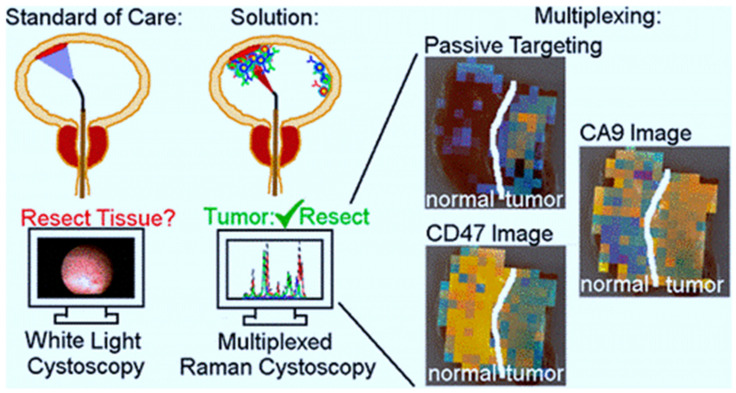
The use of surface-enhanced Raman scattering nanoparticle (NPs) for multiplexed imaging of bladder cancer tissue [83].

**Figure 2 cancers-13-02214-f002:**
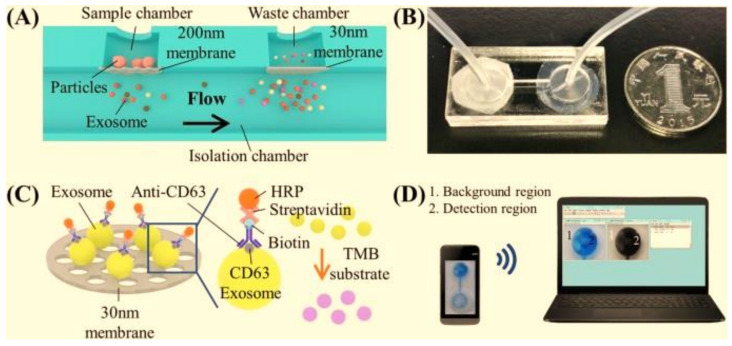
Separation and identification of EVs from urine using a microfluidic system with integrated double-filtration. (**A**) Illustration of a microfluidic system with double filtration. (**B**) Schematic of a completed double-filtration process. (**C**) Description of an on-chip direct ELISA for EV identification. (**D**) A smartphone that is used, to photograph the ELISA result, and then transported to a laptop for data processing using ImageJ [109].

**Figure 3 cancers-13-02214-f003:**
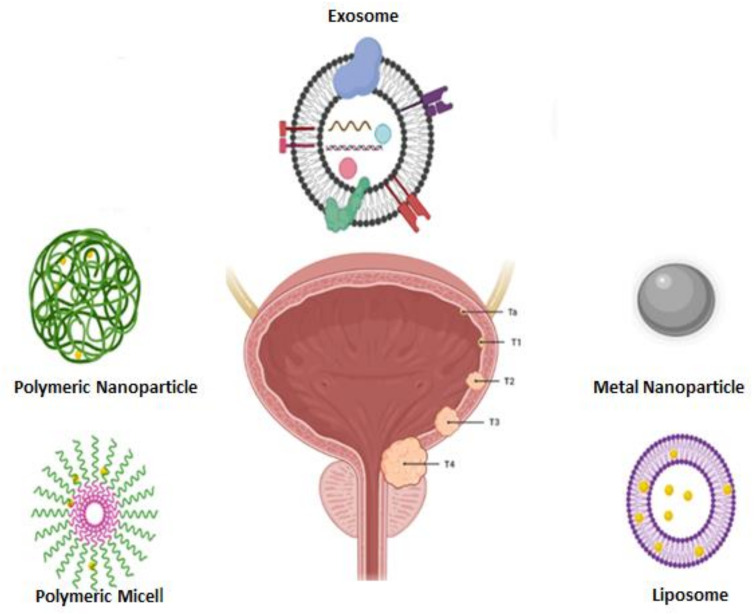
Several NPs in order to delivery of therapeutic agents in tumor site. The staging of BC is based on the situation and advance of BC cells.

**Figure 4 cancers-13-02214-f004:**
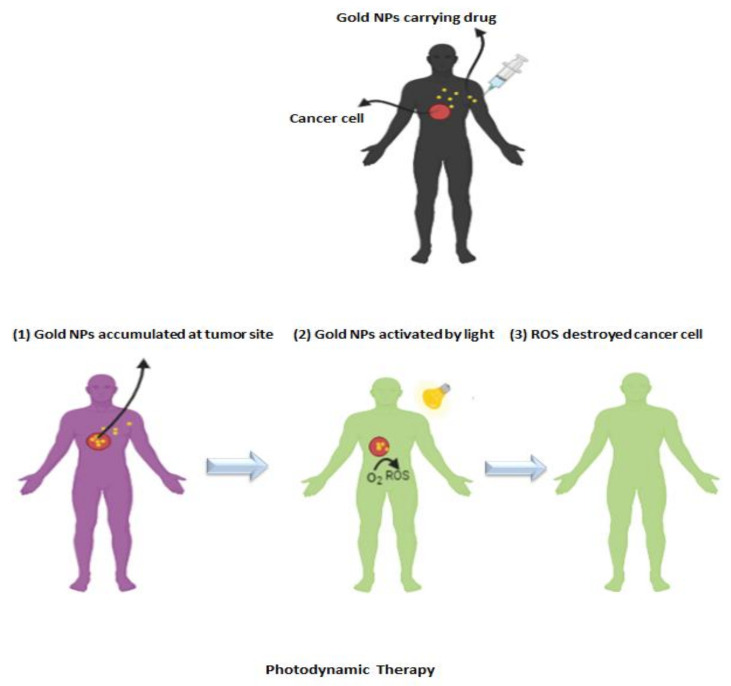
Major properties of GNPs.

**Figure 5 cancers-13-02214-f005:**
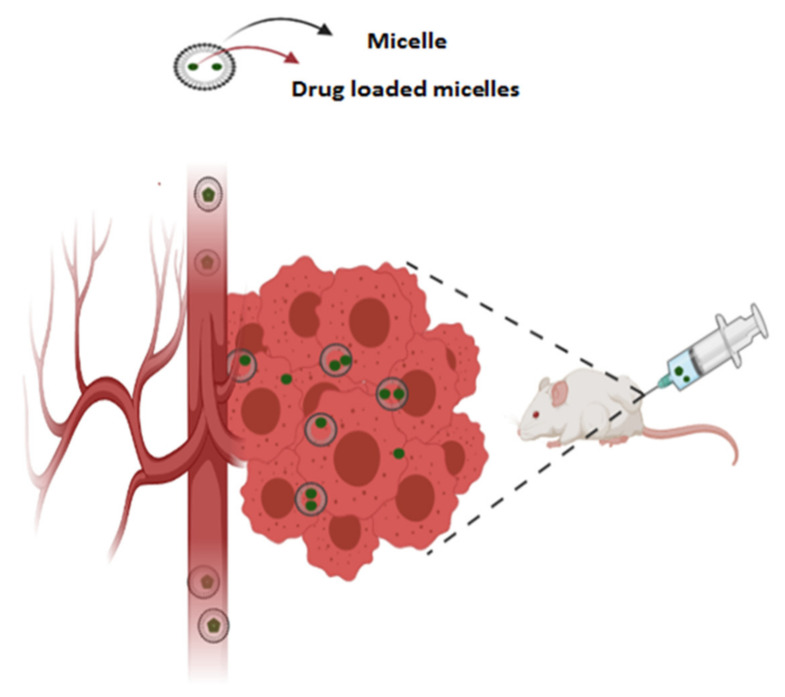
Delivery of therapeutic agent loaded in micelle.

**Figure 6 cancers-13-02214-f006:**
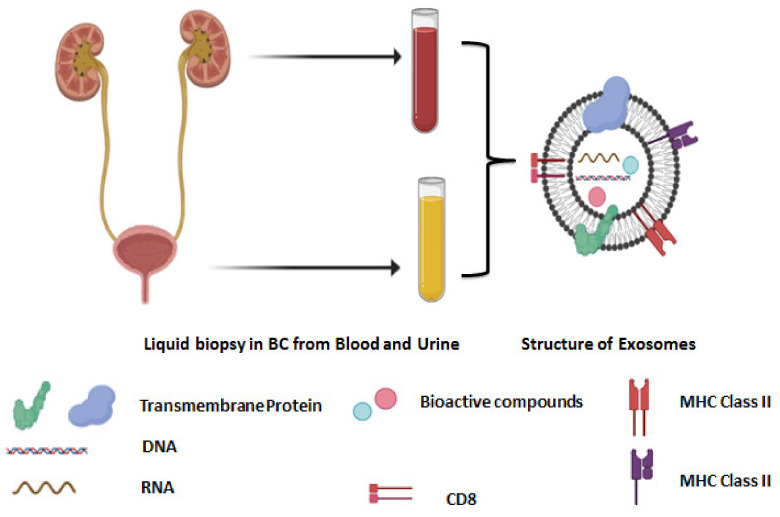
Structure of exosomes as potential source of tumor biomarkers in liquid biopsies.

**Table 1 cancers-13-02214-t001:** Applications of nanotechnology in treatment of BC [123].

Treatment Strategies	Applied NPs	Therapeutic Agents	References
Immunotherapy	Liposomes	BCG’s CWS	[133]
Targeted therapy	GNPs	Brazilian Red Propolis (BRP), survivin	[38]
	Chitosan-SPION	5-FU	[134]
	polymeric micelles	DOX, paclitaxel	[135,136]
	Polymeric NPs	gambogic acid	[137]
	Liposomes	IFN-α, DOX	[138,139]
Photo thermal therapy	Polymeric NPs	DOX and IR780	[140]

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
