# Peer review of "Nanotechnology in Bladder Cancer: Diagnosis and Treatment"

_cancers, 2021, doi:10.3390/cancers13092214_

Round 1

Reviewer 1 Report

The manuscript is very hard to read. The organization is not clear. Authors should use more full stops throughout the text to organize it better. 

The organization is too specific, to me, and it mixes general thinks with specific descriptions of nanoparticles. 

The authors either classify by type of nanoparticle or by type of application, it is very difficult to follow. 

Too much attention is paid to the clinical treatments, even though the title indicates that the review is on nanotechnology. 

I think it should be completely rewritten.

Author Response

We would like to thanks reviewer for their time and positive comments on the importance of our work. We also thank them for their constructive remarks, which we have acted upon to help further improve the manuscript.

Reviewer 2 Report

This review article describes an overview of the field of nanotechnology in bladder cancer as it pertains to diagnosis and treatment. The review is fairly comprehensive, however, there are issues that need to be addressed.

1) The manuscript will have to be proofread by somebody with English proficiency and thoroughly revised accordingly. There are several areas throughout the manuscript where poor English has been used which makes reading the manuscript complicated and often confusing. 

2) The opening statement in the abstract reads "Bladder cancer (BC) is considered a usual urinary tract malignant cancer in the world, and the mortality is almost high." This is vague and can be misconstrued. While I agree bladder cancer can be considered fairly common in men, it is not as common in women. Also, there are different types of bladder cancer, let alone different stages that effect the mortality rate. Carcinoma in situ and bladder cancer that has not spread does not have a high mortality rate. Once it spreads (regionally/distally), the mortality rate goes up considerably. This needs clarification.

3) The second sentence in the abstract reads “There are many conventional methods for diagnosis, treatment of BC such as some current biomarkers and clinical tests for BC diagnosis, radiotherapy, surgical and chemotherapy for treatment, but residual tumor cells mostly cause tumor recurrence, and also chemotherapy after transurethral resection causes high side effects and lack of the selectivity and low sensitivity in sensing.” This is one of the examples of poor English. This is an extremely long run-on sentence spliced together with several commas. This sentence describes diagnosis, treatment, reoccurrence, and side effects. This needs to be broken up into more than one sentence.

4) In section 1. Introduction, the article describes basic anatomy and development of bladder cancer. It then goes into the various treatments, diagnosis, followed by an overview of nanotechnology used for diagnosis and treatment. A suggestion would be to talk about the traditional diagnostic tools before the traditional treatments. I believe this would give the section a better flow.

5) In section 2.1. Current diagnosis approaches of bladder cancer. There is spelling error of “cystoscopy” in first sentence.

6) In section 2.2. Current biomarkers and tests in bladder cancer diagnosis (pg. 4). Au NPs are mentioned, however, Au NPs are not defined as gold nanoparticles until page 12. Au NPs should be defined when first mentioned.

7) In section 2.4. Nanomaterials in biosensor development (pg.6). There is spelling error of early in the second sentence.

8) In section 2.4. Nanomaterials in biosensor development (pg.6). GNPs should be defined as gold nanoparticles.

9) In section 2.4. Nanomaterials in biosensor development (pg.6). In the 4th paragraph, it reads graphene oxide-linked to Prussian blue (PB) (PMGO). Graphene oxide (GO) should be defined here. It is repeatedly defined at least 2 times later in the manuscript which is redundant.

10) Page 7, 2nd paragraph. What is the “222” at the beginning of the last sentence?

11) Page 8, Fig. 2 in the caption, letter D, there is spelling error of “result”.

12) Section 3. Nanomaterials in treatment of BC (page 8). The opening statements reads “As mentioned earlier, Bladder cancer (BC) is considered as one of the recurrent uro-genital cancers in the world so that 549,000 new patients are identified annually”. Bladder cancer has already been defined as BC earlier in paper (pg. 6). Also, I believe the author was trying to say “as one of the most recurrent uro-genital cancers in the world…”

13) Page 8, last paragraph. Drug delivery system should be defined as (DDS) not (DDs).

14) Section 3.2. Role of phytochemical agents in BC therapy (pg. 9). I’m not sure the reason on putting this section in. There is no mention of studies that load these potential phytochemical agents onto NPs for treatment of BC. Fucoidan, CUR, and apigenin are talked about at length but there is no reference to their use with NPs.

15) Page 21, 2nd sentence. There is an error in writing MIBC. It is written IMBC in the manuscript.

Author Response

(The authors gave the same response as above.)

Round 2

Reviewer 1 Report

Extensive changes have been applied and now everything is more clear. 

Reviewer 2 Report

Initial issues with the paper have been addressed.